# Antifungal Activity against *Botrytis cinerea* of 2,6-Dimethoxy-4-(phenylimino)cyclohexa-2,5-dienone Derivatives

**DOI:** 10.3390/molecules24040706

**Published:** 2019-02-15

**Authors:** Paulo Castro, Leonora Mendoza, Claudio Vásquez, Paz Cornejo Pereira, Freddy Navarro, Karin Lizama, Rocío Santander, Milena Cotoras

**Affiliations:** 1Laboratorio de Micología, Facultad de Química y Biología, Universidad de Santiago de Chile, Avenida Libertador Bernardo O’Higgins 3363, Santiago 518000, Chile; leonora.mendoza@usach.cl (L.M.); paz.cornejo@usach.cl (P.C.P.); freddy.navarro@usach.cl (F.N.); karin.lizama@usach.cl (K.L.); milena.cotoras@usach.cl (M.C.); 2Laboratorio de Microbiología Molecular, Departamento de Biología, Facultad de Química y Biología, Universidad de Santiago de Chile, Santiago 518000, Chile; claudio.vasquez@usach.cl; 3Departamento de Ciencias del Ambiente, Facultad de Química y Biología, Universidad de Santiago de Chile, Casilla 40 Correo 33, Santiago 518000, Chile; rocio.santanderm@usach.cl

**Keywords:** *Botrytis cinerea*, antifungal activity, laccase, 2,6-dimethoxy-4-(phenylimino)cyclohexa-2,5-dienone derivatives

## Abstract

In this work the enzyme laccase from *Trametes versicolor* was used to synthetize 2,6-dimethoxy-4-(phenylimino)cyclohexa-2,5-dienone derivatives. Ten products with different substitutions in the aromatic ring were synthetized and characterized using ^1^H- and ^13^C-NMR and mass spectrometry. The 3,5-dichlorinated compound showed highest antifungal activity against the phytopathogen *Botrytis cinerea*, while the *p*-methoxylated compound had the lowest activity; however, the antifungal activity of the products was higher than the activity of the substrates of the reactions. Finally, the results suggested that these compounds produced damage in the fungal cell wall.

## 1. Introduction

*Botrytis cinerea* is a phytopathogenic fungus promoted by the presence of free surface water or high relative humidity and causing significant crop losses in a wide variety of plant species [1]. Regarding the control, methods aiming to reduce humidity can be combined to help decrease this disease, in addition to chemical fungicides or biocontrol treatments [1]. Chemical control is the most common way to manage *B. cinerea*, mainly using synthetic compounds [1]. The restriction of this type of control becomes necessary to reduce the impact on the environment [2] and to avoid the acquired resistance to botrycides [3,4,5,6,7]. For this reason, the development of new antifungal compounds is essential. Natural products can be a good alternative to commercial fungicides [8,9]. For instance, phenolic compounds, terpenoids, nitrogen-containing compounds, and aliphatic compounds isolated from plants have shown antifungal activities [10,11,12]. Additionally, new antifungal compounds against *B. cinerea* derived of natural products have been synthesized, such as derivatives of natural stilbene resveratrol [13], chlorophenyl derivatives [14], or different clovanes [15].

Several phenolic metabolites found in grape pomace have shown low antifungal activity against *B. cinerea* [16], therefore, it is possible to increase the biological activity of phenolic compounds using the enzyme laccase [17]. These enzymes (benzenediol: oxygen oxidoreductase, EC 1.10.3.2) belong to the oxidase group, and they are also used for cleaner industrial application [18]. Laccases are also known as multicopper oxidases, they belong to the family of copper-containing phenol oxidases [19] and can oxidize a diversity of compounds, e.g., phenolic and nonphenolic compounds [18]. Aromatic compounds can produce reactive radical intermediates, which undergo self-coupling reactions, thus forming different dimers and trimers [20,21,22,23,24]. This enzyme has been previously used to improve the activity of antibiotics [25,26]. On the other hand, the synthesis of a heterodimeric compound (2,6-dimethoxy-4-(phenylimino)cyclohexa-2,5-dienone) by the laccase-mediated coupling reaction between syringic acid and aniline was reported, this compound showed an antifungal effect against *B. cinerea* with an EC_50_ value of 0.14 mM [27]. 

Antifungal compounds have shown several inhibition mechanisms related to the molecular structure. For instance, the resveratrol derivative (*E*)-3,5-dimethoxy-β-(2-furyl)-styrene cause cell membrane damage against *B. cinerea* [13]. Phenylpyrroles induce morphological alterations of germ tubes [28]. Fungicides such as dinocap and fuazinam have been described as uncouplers of oxidative phosphorylation [29,30] and fungicides, like dicloran, cloroneb, and etazol, affect cell wall synthesis [28]. 

This work aimed to determine the antifungal activity against *B. cinerea* and the effect on the cell wall integrity of ten 2,6-dimethoxy-4-(phenylimino)cyclohexa-2,5-dienone derivatives (**3a–j**) obtained by reaction of syringic acid (**1**) with substituted anilines (**2a–j**). To analyze the effect of the carboxylic group in these laccase-catalyzed reactions, syringaldehyde was used instead syringic acid and the reaction product was characterized.

## 2. Results and Discussion

### 2.1. Laccase-Mediated Synthesis of 2,6-Dimethoxy-4-(phenylimino)cyclohexa-2,5-dienone Derivatives

In this work, laccase catalyzed reactions using **1** and **2a–j** were carried out. It has been previously reported that using laccases from different fungal sources (*Trametes* sp. and *Rhizoctonia praticola*), catalyze reactions between phenolic compounds and anilines, heterodimeric compounds are formed [25,26,31,32], similarly found in this work (Scheme 1).

To determine the reaction yields in the formation of the products, different substrate ratios were analyzed. Excluding reactions 1, 7 and 10, most reactions reached higher yields using ratio 1:1 (syringic acid:aniline) (Table 1). Moreover, when aniline was used as substrate, the same result was reported [27], indicating that the increase of the concentration of one of them decrease the yield of the obtained compounds.

Highest yields were obtained using 3-chloroaniline and 3,5-dichloroaniline as substrates (**2b** and **2h**) (Table 1). This high yield could be explained because the oxidation by laccase (from *Trametes versicolor*) of 3-chloroaniline does not occur [33]. On the other hand, using methoxyanilines (**2c** and **2d**) low yields were obtained, due to a high amount of side products (data not shown).

On the other hand, yield did not increase when the enzyme concentration was increased (data not shown). Bollag et al. [31] showed that the prolonged incubation or higher enzyme amounts caused further polymerization reaction decreasing cross-coupling formation. Furthermore, Itoh et al. [34] concluded that reactivity of laccase mediated reaction between phenolic acids and chlorophenols is due to the substrate specificity of the laccase rather than the chemical property of the substrates, which could explain the lack of relations among electron donating and withdrawing groups and yield of the reactions.

The ten synthetized compounds were purified using semipreparative chromatography and were identified (Figure 1) using ^1^H-NMR and ^13^C-NMR spectra and mass spectrometry (Appendix A). Compound **3b** showed two aliphatic proton signals (δ 3.670 (s, 3H, H8) and δ 3.874 (s, 3H, H7)) and two aliphatic carbon signals (δ 56.212 and δ 56.321) that determined the presence of two methoxy moieties. Two olefin hydrogen signals at higher fields (δ 6.010 (d, 1H, H3 *J* = 1.9 Hz) and δ 6.368 (d, 1H, H5 *J* = 1.9 Hz)), the olefin carbon signals (δ 98.583 (C3) and δ 111.717 (C5)) and one carbon signal at δ 176.633 (C1) indicated the quinonoid character of the products. Table 2 presents the NMR data (chemical shift assignments for short and long-range heteronuclear coupling) of compound **3b**. 

The only difference in spectra signals between compounds **3a** and **3b** (Appendix A) was in the aromatic region of the spectra. Compound **3b** showed four aromatic proton signals δ 6.737 (d, 1H, H6’, *J* = 8.0 Hz), δ 6.888 (s, 1H, H2’), δ 7.159 (d, 1H, H4’, *J* = 8.0 Hz), and δ 7.316 (t, 1H, H5’, *J* = 8.0 Hz) that indicated the existence of a *m*-substituted aromatic fragment. The assignation of the entire molecule was achieved by using two-dimensional NMR analysis. Therefore, identifying this compound as 4-(3’-chlorophenylimino)-2,6-dimethoxycyclohexa-2,5-dienone (Figure 1). 

The spectra of compounds **3c–j** (Appendix A) only showed differences in the aromatic region; the assignment of the ^1^H and ^13^C-NMR spectra can be found in the spectroscopic data section (Section 3.4). Figure 1 shows the structures of the ten synthetized compounds in this work. To our knowledge compounds **3a** and **3f** were previously synthesized [31], ^1^H-NMR spectra for compounds **3a** and **3f** (spectroscopic data Section 3.4) have the same number of signals and comparable chemical shifts and coupling constants like those found by Bollag et al. [31]; furthermore, the mass spectra of **3a** and **3f** showed a base peak with *m*/*z* 277 and 311, respectively, corresponding to the molecular ions, equivalent to the previously described data [31]. Therefore, the other eight compounds (**3b**, **3c**, **3d**, **3e**, **3g**, **3h**, **3i** and **3j**) have not been previously reported.

Interesting, compound **3a** was also obtained using syringaldehyde instead of syringic acid in the reaction with 4-chloroaniline. This could be explained with an extra step when using syringaldehyde, an oxidation of the aldehyde to a carboxylic acid (syringic acid), similar oxidations has been previously described using several aromatic aldehydes with laccase, yielding carboxylic acids [35]. Hence, syringaldehyde is oxidized to syringic acid and then the same product (compound **3a**) could be found in both reactions, starting with syringaldehyde or with syringic acid. However, this synthesis had a very low yield (data not shown).

### 2.2. Antifungal Activity

Antifungal activity of compounds **1** and **2a–j** and compounds **3a–j** against *B. cinerea* was measured on mycelial growth in solid media and the EC_50_ were calculated using the mycelial growth (Table 3 and Table 4). The most active compound was the 3,5-dichloro-substituted product (compound **3h**), while compound **3c** had the lowest activity. It has been reported that the substituent affects the antifungal activity of a molecule [36], for instance, the position of the chlorine atom in the aromatic ring is important for the antifungal activity against *B. cinerea* since para-substituted compound (**3a**) and ortho-substituted compound (**3j**) were more active than the meta-substituted compound and unsubstituted compound (EC_50_ = 0.14 ± 0.02) [27], while activity of meta-substituted compound (compound **3b**) and unsubstituted compound are similar [27]. The number of chlorine atoms in the aromatic ring is also important, both dichlorinated compounds **3f** and **3h** showed higher antifungal activity than mono chlorinated compounds **3a**, **3b**, and **3j**, however, dichlorinated compound **3i** showed an antifungal activity comparable to the monochlorinated compounds, therefore, the number and position of chlorine atoms in the aromatic ring seems to be important for the antifungal activity of these compounds. 

Furthermore, the methoxy derivative compounds (**3c** and **3d**) were less active against the fungus than the other compounds, even the nonsubstituted compound **3** [27], the same effect was observed for aspirin derivatives, where the methoxy para-substituted derivative showed almost 30% less antifungal activity against *B. cinerea* than the chlorinated para-substituted compound [37]. Similar behavior was previously reported for oxadiazole derivatives when tested the activity of the methoxy meta-substituted oxadiazole derivative against *B. cinerea*, and its activity was less than half compared to the nonsubstituted compound [38]. Usually, the chloro-substituted compounds have higher antifungal activity in commercial fungicides, for example, chlorine compounds like boscalid, chlorothalonil, and iprodione have been used to control *B. cinerea* [6]. The antifungal activity of iprodione has been tested against this strain of *B. cinerea*, showing an EC_50_ of 0.015 ± 0.003 mM [27], this antifungal activity is in the same order of magnitude than the most active compound obtained in this work (**3h**). Additionally, *p*-nitro and *p*-trifluoromethyl compounds (**3e** and **3g**) were tested against this fungus, **3e** showed no antifungal activity, probably because of the low solubility of this molecule, for this reason **3e** was not used in further assays. Compound **3g** only showed an intermediate antifungal activity compared to the rest of the synthetized molecules in this work. Lastly, most of the substrates used in the reactions (i.e., **1** and **2a–j**) showed lower antifungal activity than the products (Table 4), only **2f** was more active than **3f**.

### 2.3. Effect on the Cell Wall Integrity of B. cinerea

To analyze the effect of the compounds on the cell wall integrity, the dye calcofluor white (CFW) was used. This dye binds to β-1,3 and β-1,4 polysaccharides, for example chitin, which is a primary component of the cell wall in fungi, and fluorescence of the hyphae can be detected [39]. Figure 2 shows the effect of compound **3a** on the cell wall of *B. cinerea*. Treatment with this compound showed lower fluorescence intensity than the negative control (acetone), indicating that this compound can damage the cell wall of this fungus. The same assay was performed using compounds **3b–j**. The ten synthesized compounds caused a decrease of the fluorescence intensity compared to the control; relative fluorescence intensity is observed in Figure 3. This result could be attributed to the toxicity of quinones, which could be connected to the production of reactive oxygen species (ROS) which cause oxidation of cell molecules [40]. Quinone derivative *N*-acetyl-*p*-benzoquinone imine (NAPQI) can react with nucleophiles such as thiol groups of proteins or glutathione [41,42]; this last molecule is an important antioxidant molecule in fungi [43]. On the other hand, some aromatic antifungal compounds have shown effects on cell wall synthesis [6] by inhibiting chitin and glucan synthases [44], enzymes that catalyze the synthesis of the main polymers of the cell wall in fungi.

## 3. Materials and Methods

### 3.1. General Experimental Procedures

The NMR spectra of **3a–j** were acquired using a Bruker Avance 400 MHz spectrometer (Bruker, Billerica, MA, USA) (400,133 MHz for ^1^H, 100.624 MHz for ^13^C). Measurements were done in CDCl_3_ at 27 °C. Chemical shifts were calibrated to solvent signal: CHCl_3_ 7.26 ppm (residual signal solvent) and 77.16 ppm for ^1^H and ^13^C, respectively, and informed relative to Me_4_Si. Thin-layer chromatography was done with a Merck Kiesegel 60 F_254_, 0.2 mm thick and semipreparative thin layer chromatography on Merck Kieselgel 60 F_254_ 0.25 mm thick. A Thermo Scientific GC-MS system (GC: model: Trace 1300 and MS: model TSQ8000Evo) (Waltham, Massachusetts, USA) was used to analyze the sample. The separation was performed on a 60 m × 0.25 mm internal diameter fused silica capillary column coated with 0.25 μm film Rtx-5MS. The oven temperature was maintained at 40 °C for 5 min, then it was programmed from 40 to 80 at 5 °C/min for 1 min, then from 80 to 300 at 10 °C/min and finally maintained at 300 °C for 10 min. The mode used was splitless injection, helium was used as carrier gas, and flow-rate was 1.2 mL/min. Mass spectra were recorded over a range of 40 to 400 atomic mass units at 0.2 s/scan. Solvent cut time was 11 min. Ionization energy was 70 eV.

### 3.2. Chemical Reagents

Laccase from *Trametes versicolor* (EC 1.10.3.2), lysing enzymes from *Trichoderma harzianum*, Calcofluor white stain, 4-hydroxy-3,5-dimethoxy-benzoic acid (syringic acid), 3,5-dimethoxy-4-hydroxybenzaldehyde (syringaldehyde), 4-chloroaniline, 2,5-dichloroaniline, 3,5-dichloroaniline, and 4-nitroaniline were obtained from Sigma Chemical Co. (St. Louis, MO, USA). 3-chloroaniline, 4-methoxyaniline, 3-methoxyaniline, 3,4-dichloroaniline, 2-chloroaniline, organic solvents, and salts were obtained from Merck (Hohenbrunn, Germany). 4-(trifluoromethyl)aniline was obtained from Santa Cruz Biotechnology (Finnell St, Dallas Tx).Agar was obtained from Difco Laboratories (Detroit, MI, USA).

### 3.3. Laccase-Mediated Synthesis of 2,6-Dimethoxy-4-(phenylimino)cyclohexa-2,5-dienone Derivatives (Compounds ***3a–j***)

Syringic acid with an aniline derivative at different ratios (1:1, 1:2, and 2:1) (e.g., for ratio 1:1 means 0.1 mmol for both syringic acid and the aniline derivative were used) and different enzyme quantities (2.25, 4.5, and 9 U) were tested to increase the yield of the synthesized compounds.

For the first reaction, syringic acid (**1**) and 4-chloroaniline (**2a**) were dissolved in 1 mL ethyl acetate and laccase was dissolved in 1 mL sodium acetate buffer (20 mM, pH 4.5). Both solutions were mixed and stirred at 100 rpm for 180 min at 22 °C. Afterwards, the solvent was evaporated at 40 °C using a rotary evaporator. The synthetized compounds were purified by using semipreparative thin layer chromatography with hexane: ethyl acetate (1:1) as an eluent system. Same procedure was carried out using a different substituted aniline (**2b–j**).

Alternatively, compound **3a** was also found when using syringaldehyde and 4-chloroaniline under the same conditions described above.

### 3.4. Spectroscopic Data

Compound **3a**
*(4-(4′-chlorophenylimino)-2,6-dimethoxycyclohexa-2,5-dienone)* Yield 55.5%. ^1^H-NMR (CDCl_3_, 400 MHz) δ 3.668 (s, 3H, H8), 3.870 (s, 3H, H7), 6.040 (d, 1H, *J* = 1.9 Hz, H3), 6.377 (d, 1H, *J* = 1.9 Hz, H5), 6.816 (d, 2H, *J* = 8.5 Hz, H2’), 7.361(d, 2H, *J* = 8.5 Hz, H3’); ^13^C-NMR (CDCl_3_, 100 MHz) δ 56.188 (C8), 56.298 (C7), 98.529 (C3), 111.846 (C5), 122.040 (C2’), 129.333 (C3’), 130.749 (C4’), 148.777 (C1’), 154.813 (C6), 155.745 (C2), 157.642 (C4), 176.647 (C1). mp 208.0–209.1 °C. GC-MS RI_(Rtx-5ms)_ = 2366, C_14_H_12_O_3_NCl EI-MS *m*/*z*: 111 (15); 150 (16); 178 (15); 182 (17); 197 (35); 212 (15); 224 (22); [M]^+^ = 277 (100); [M + 1]^+^ = 278 (17); [M + 2]^+^ = 279 (36).

Compound **3b** (*4-(3′-chlorophenylimino)-2,6-dimethoxycyclohexa-2,5-dienone)* Yield 72.0%. ^1^H-NMR (CDCl_3_, 400 MHz) δ 3.670 (s, 3H, H8), 3.874 (s, 3H, H7), 6.010 (d, 1H, *J* = 1.9 Hz, H3), 6.368 (d, 1H, *J* = 1.9 Hz, H5), 6.737 (d, 1H, *J* = 7.9 Hz, H6’), 6.888 (s, 1H, *J* = 8.0 Hz, H2’), 7.159 (d, 1H, *J* = 8,0 Hz, H4’), 7.316 (t, 1H, *J* = 7.9 Hz, H5’); ^13^C-NMR (CDCl_3_, 100 MHz) δ 56.212 (C8), 56.321 (C7), 98.583 (C3), 111.717 (C5), 118.705 (C6’), 120.635 (C2’), 125.023 (C4’), 130.247 (C5’), 134.928 (C3’), 151.483 (C1’), 154.881 (C6), 155.741 (C2), 157.816 (C4), 176.633 (C1). mp 155.7–156.0 °C. GC-MS RI_(Rtx-5ms)_ = 2340, C_14_H_12_O_3_NCl EI-MS m/z: 69 (22); 75 (52); 111 (58); 113 (21); 140 (20); 178 (26); 182 (24); 197 (43) [M]^+^ = 277 (100); [M + 1]^+^ = 278 (17); [M + 2]^+^ =279 (36).

Compound **3c**
*(4-(4′-methoxyphenylimino)-2,6-dimethoxycyclohexa-2,5-dienone)* Yield 23.7%. ^1^H-NMR (CDCl_3_, 400 MHz) δ 3.691 (s, 3H, H8), 3.846 (s, 3H, H7’), 3.870 (s, 3H, H7), 6.242 (d, 1H, *J* = 1.9 Hz, H3), 6.467 (d, 1H, *J* = 1.9 Hz, H5), 6.903 (d, 2H, *J* = 8.9 Hz, H2’), 6.956 (d, 2H, *J* = 8.9 Hz, H3’); ^13^C-NMR (CDCl_3_, 100 MHz) δ 55.656 (C7’), 56.104 (C8), 56.277 (C7), 98.966 (C3), 112.113 (C5), 114.594 (C3’), 123.047 (C2’), 143.196 (C1’), 154.661 (C6), 155.739 (C2), 156.809 (C4), 158.018 (C4’), 176.669 (C1). mp 111.7–112.3 °C. GC-MS RI_(Rtx-5ms)_ = 2451, C_15_H_15_O_4_N EI-MS *m*/*z*: 134 (10); 172 (20); 198 (12); 200 (9); 212 (9); 230 (30); 240 (12); 258 (45); [M]^+^ = 273 (100); [M + 1]^+^ = 274 (18).

Compound **3d**
*(2,6-dimethoxy-4-(3′-methoxyphenylimino)cyclohexa-2,5-dienone)* Yield 24.24%. ^1^H-NMR (CDCl_3_, 400 MHz) δ 3.656 (s, 3H, H8), 3.819 (s, 3H, H7’), 3.868 (s, 3H, H7), 6.125 (d, 1H, *J* = 2.0 Hz, H3), 6.392 (d, 1H, *J* = 2.0 Hz, H5), 6.429 (d, 1H, *J* = 7.7 Hz, H6’), 6.452 (s, 1H, H2’), 6.736 (d, 1H, *J* = 8.1 Hz, H4’), 7.281 (t, 1H, *J* = 8.0 Hz, H5’); ^13^C-NMR (CDCl_3_, 100 MHz) δ 55.476 (C7’), 56.124 (C8), 56.248 (C7), 99.051 (C3), 106.387 (C2’), 110.942 (C4’), 112.023 (C5), 112.868 (C6’), 126.946 (C5’), 151.664 (C1’), 154.783 (C6), 155.569 (C2), 157.264 (C4), 160.402 (C3’), 176.785 (C1). mp 133.5–134.6 °C. GC-MS RI_(Rtx-5ms)_ = 2403, C_15_H_15_O_4_N EI-MS *m*/*z*: 159 (13); 187 (16); 199 (19); 200 (13); 212 (12); 215 (22); 230 (21); 242 (24) [M]^+^ = 273 (100); [M + 1]^+^ = 274 (18).

Compound **3e**
*(2,6-dimethoxy-4-(4′-nitrophenylimino)cyclohexa-2,5-dienone)* Yield 56.8%. ^1^H-NMR (CDCl_3_, 400 MHz) δ 3.660 (s, 3H, H8), 3.895 (s, 3H, H7), 5.845 (d, 1H, *J* = 1.8 Hz, H3), 6.363 (d, 1H, *J* = 1.8 Hz, H5), 6.959 (d, 2H, *J* = 8.8 Hz, H2’), 8.279(d, 2H, *J* = 8.8 Hz, H3’); ^13^C-NMR (CDCl_3_, 100 MHz) δ 56.341 (C8), 56.439 (C7), 98.339 (C3), 111.172 (C5), 120.620 (C2’), 125.205 (C3’), 144.864 (C4’), 155.257 (C1’), 156.102 (C6), 156.170 (C2), 157.940 (C4), 176.321 (C1). mp 208.0–209.1 °C. GC-MS RI_(Rtx-5ms)_ = 2690, C_14_H_12_O_5_N_2_ EI-MS *m*/*z*: 16 (33); 128 (21); 143 (29); 156 (25); 168 (18); 169 (20); 197 (38); 211 (26); [M]^+^ = 288 (100); [M + 1]^+^ = 289 (16).

Compound **3f**
*(4-(3′,4′-dichlorophenylimino)-2,6-dimethoxycyclohexa-2,5-dienone)* Yield 50.4%. ^1^H-NMR (CDCl_3_, 400 MHz) δ 3.690 (s, 3H, H8), 3.874 (s, 3H, H7), 5.982 (s, 1H, H3), 6.347 (s, 1H, H5), 6.715 (d, 1H, *J*= 8.4 Hz, H6’), 6.994 (s, 1H, H2’), 7.449 (d, 1H, *J* = 8.4 Hz, H5’); ^13^C-NMR (CDCl_3_, 100 MHz) δ 56.319 (C8), 56.355 (C7), 98.322 (C3), 111.572 (C5), 120.097 (C6’), 122.367 (C2’), 128.704 (C3’), 130.913 (C5’), 133.211 (C4’), 149.770 (C1’), 155.033 (C6), 155.973 (C2), 158.242 (C4), 176.486 (C1). mp 160.1–161.5 °C. GC-MS RI_(Rtx-5ms)_ = 2556, C_14_H_11_O_3_NCl_2_ EI-MS m/z: 109 (22); 145 (18); 184 (21); 212 (20); 216 (26); 231 (41); 233 (28); 258 (24); [M]^+^ = 311 (100); [M + 2]^+^ = 313 (65); [M + 4]^+^ = 315 (13).

Compound **3g**
*(4-(4′-(trifluoromethyl)phenylimino)-2,6-dimethoxycyclohexa-2,5-dienone)* Yield 12.7%. ^1^H-NMR (CDCl_3_, 400 MHz) δ 3.643 (s, 3H, H8), 3.868 (s, 3H, H7), 5.928 (s, 1H, H3), 6.367 (s, 1H, H5), 6.933 (d, 2H, *J* = 8.1 Hz, H2’), 7.633 (d, 2H, *J* = 8.1 Hz, H3’); ^13^C-NMR (CDCl_3_, 100 MHz) δ 56.211 (C8), 56.302 (C7), 98.439 (C3), 111.559 (C5), 120.453 (C2’), 124.319 (q, *J* = 271.6 Hz, C5’), 126.425 (q, *J* = 3.7 Hz, C3’), 126.936 (q, *J* = 32.7 Hz, C4’), 153.272 (C1’), 154.987 (C6), 155.898 (C2), 157.775 (C4), 176.491 (C1). mp 130–133 °C. GC-MS RI_(Rtx-5ms)_ = 2084, C_15_H_12_O_3_NF_3_ EI-MS *m*/*z*: 53 (22); 69 (43); 95 (37); 125 (30); 145 (96); 184 (29); 197 (57); 212 (52); 221 (22); [M]^+^ = 311 (100).

Compound **3h**
*(4-(3′,5′-dichlorophenylimino)-2,6-dimethoxycyclohexa-2,5-dienone)* Yield 74.0%. ^1^H-NMR (CDCl_3_, 400 MHz) δ 3.699 (s, 3H, H8), 3.874 (s, 3H, H7), 5.935 (d, 1H, *J* = 2.1 Hz, H3), 6.327 (d, 1H, *J* = 2.1 Hz, H5), 6.758 (d, 2H, *J* = 1.8 Hz, H2’), 7.167 (d, 1H, *J* = 1.8 Hz, H4’); ^13^C-NMR (CDCl_3_, 100 MHz) δ 56.375 (C8 and C7), 98.324 (C3), 111.375 (C5), 118.863 (C2’), 124.712 (C4’), 135.532 (C3’ and C5’), 152.147 (C1’), 155.044 (C6), 155.938 (C2), 158.388 (C4), 176.473 (C1). mp. 170–173 °C. GC-MS RI_(Rtx-5ms)_ = 2472, C_14_H_11_O_3_NCl_2_ EI-MS *m*/*z*: 109 (18); 145 (19); 212 (18); 216 (19); 231 (29); 233 (22); 246 (14); [M]^+^ = 311 (100); [M + 2]^+^ = 313 (66); [M + 4]^+^ = 315 (13).

Compound **3i**
*(4-(2′,5′-dichlorophenylimino)-2,6-dimethoxycyclohexa-2,5-dienone)* Yield 38.3%. ^1^H-NMR (CDCl_3_, 400 MHz) δ 3.678 (s, 3H, H8), 3.893 (s, 3H, H7), 5.794 (d, 1H, *J* = 1.8 Hz, H3), 6.416 (d, 1H, *J* = 1.8 Hz, H5), 6.838 (d, 1H, *J* = 2.2 Hz, H3’), 7.094 (dd, 1H, *J* = 8.6, 2.2 Hz, H5’), 7.382 (d, 1H, *J* = 8.6 Hz, H6’); ^13^C-NMR (CDCl_3_, 100 MHz) δ 56.418 (C8 and C7), 98.759 (C3), 111.205 (C5), 121.133 (C3’), 123.120 (C1’), 125.644 (C5’), 131.130 (C6’), 133.044 (C4’), 148.282 (C2’), 155.162 (C6), 155.840 (C2), 159.208 (C4), 176.453 (C1). Mp. 198–202 °C. GC-MS RI_(Rtx-5ms)_ = 2449, C_14_H_11_O_3_NCl_2_ EI-MS *m*/*z*: 190 (24); 212 (25); 231 (29); 233 (79); 261 (25); 276 (92); 278 (29); [M]^+^ = 311 (100); [M + 2]^+^ = 313 (73); [M + 4]^+^ = 315 (15).

Compound **3j**
*(4-(2′-chlorophenylimino)-2,6-dimethoxycyclohexa-2,5-dienone)* Yield 44.9%. ^1^H-NMR (CDCl_3_, 400 MHz) δ 3.637 (s, 3H, H8), 3.887 (s, 3H, H7), 5.862 (s, 1H, H3), 6.460 (s, 1H, H5), 6.797 (d, 1H, *J* = 7.7 Hz, H6’), 7.119 (t, 1H, *J* = 7.6 Hz, H5’), 7.274 (t, 1H, *J* = 7.6 Hz, H4’), 7.458 (d, 1H, *J* = 8.0 Hz, H3’); ^13^C-NMR (CDCl_3_, 100 MHz) δ 56.158 (C8), 56.332 (C7), 99.002 (C3), 111.548 (C5), 121.317 (C6’), 124.984 (C1’), 125.949 (C5’), 127.326 (C4’), 130.277 (C3’), 147.388 (C2’), 154.971 (C6), 155.618 (C2), 158.576 (C4), 176.616 (C1). mp 143–146 °C. GC-MS RI_(Rtx-5ms)_ = 2449, C_14_H_12_O_3_NCl EI-MS *m*/*z*: 150 (17); 170 (16); 178 (21); 197 (25); 199 (54); 214 (14); 242 (54); [M]^+^ = 277 (100); [M + 1]^+^ = 278 (16); [M + 2]^+^ = 279 (36).

### 3.5. Fungal Strain and Culture Conditions

The strain G29 of *B. cinerea* used in this work was isolated from infected grapes (*Vitis vinifera*) and genetically characterized by the INIA, La Platina, Chile [45]. It was kept on malt yeast extract agar slants with 0.2% (*w*/*v*) yeast extract, 2% (*w*/*v*) malt extract, and 1.5% (*w*/*v*) agar) at 4 °C. For the cell wall integrity assay, liquid minimal medium of pH 6.5 was used, containing KH_2_PO_4_ (1 g/L), MgSO_4_·7H_2_O (0.5 g/L), KCl (0.5 g/L), K_2_HPO_4_ (0.5 g/L), FeSO_4_·7H_2_O (0.01 g/L),1% (*w*/*v*) glucose as a carbon source, and 4.6 g/L ammonium tartrate as a nitrogen source.

### 3.6. Antifungal Assay

#### Effect on Mycelial Growth

The antifungal activity of the compounds was evaluated in vitro as described by Caruso et al. [13]. Compounds were dissolved in acetone and then added to Petri dishes along with malt yeast agar medium. Inhibition percentages were calculated after 72 h of incubation. Antifungal activity was expressed as the concentration that reduced mycelial growth by 50% (EC_50_), calculated by regressing the antifungal activity percentage against compound concentration. These experiments were done at least in triplicate.

### 3.7. Effect on the Cell Wall Integrity of B. cinerea

The effect of compounds **3a–j** on cell wall integrity was evaluated using the method described by Mendoza et al. [27]. Compounds **3a–j** were tested at 0.16 mM. To measure the effect of these compounds on the cell wall, fluorescence intensity was quantified using ImageJ (v1.80), an outline was drawn around each hypha, and mean fluorescence was measured, along with several adjacent background readings. Mean fluorescence was compared to the negative control (maximum fluorescence).

## 4. Conclusions

Ten compounds were synthetized; two of them (compounds **3a** and **3f**) have been previously described. All the products showed higher antifungal activity than the substrates. Chloro-substituted compounds showed the highest antifungal effect against *B. cinerea* being the 3,5-dichlorinated product **3h** the most active. Synthesis using syringic acid or syringaldehyde with *p*-chloroaniline yield the same main product (compound **3a**). Finally, regarding the inhibition mechanism of these compounds, the results suggest that these compounds damage the cell wall.

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
