# Peer review of "Antifungal Activity against Botrytis cinerea of 2,6-Dimethoxy-4-(phenylimino)cyclohexa-2,5-dienone Derivatives"

_molecules, 2019, doi:10.3390/molecules24040706_

Round 1

Reviewer 1 Report

Castro and co-workers reported the antifungal activity against Botrytis cinerea of 2,6-dimethoxy-4-(phenylimino)benzenone derivatives.

This paper is too similar to that previously reported by the same authors in reference 27 (J. Chil. Chem. Soc. 2016, 61, 3039-3042). Most of the experimental/chemical details and biological assays reported here, were previously established in ref. 27. Moreover, several concluding remarks here are the same of the ref. 27. Just three of the four synthesized compounds are new because compound 1 was previously reported; however, the synthetic methodology used is exactly the same than that established in reference 27, for that there is not novelty.

Other comments.

-Several style/grammatical corrections must be undertaken. Please see the attached document.

- References also must be revised and corrected.

- A re-numbering of the obtained compounds should be made.

- Systematic names for the obtained compounds should be assigned.

- Some generalities made about the observed activities for compounds 1-4 are hard to assure due to just four examples  is a too short set of samples for a SAR analyses. Please be careful. Perhaps, more examples are required for a more reliable analysis.

- A commercial antibiotic is required to be used as reference for comparison and analysis of the antifungal assays.

- The spectroscopic characterization of the obtained products should be revised. Mainly, the mass spectra require revision. This referee recommend the attaching of the NMR and mass spectra for all products as Supporting Information.

- The procedure for calculate the reaction yields must be revised. This referee consider that it is wrong. Moreover, some yields reported in the experimental section does not matches with those reported in Table 1.

Due that there is not novelty in (synthetic and biological) methodologies, perhaps supplying several new examples and their antifungal results could increase the interest on this paper.

Author Response

Response to Reviewer 1 Comments

Point 1: Several style/grammatical corrections must be undertaken. Please see the attached document.

Response 1: Attached document was used to correct style and grammatical errors in the whole manuscript.

Point 2: References also must be revised and corrected.

Response 2: References were revised and corrected

Point 3: A re-numbering of the obtained compounds should be made.

Response 3: Obtained compounds were re-numbered (3a to 3j) (ten compounds). For example, compound 1 is now 3a.

Point 4: Systematic names for the obtained compounds should be assigned

Response 4:  In the first version of the manuscript, the obtained compounds were called 2,6-dimethoxy-4-(phenylimino)benzenone derivatives. The compound names were corrected and now they are called 4-(phenylimino)-2,6-dimethoxycyclohexa-2,5-dienone derivatives (highlighted in yellow in the manuscript)

Point 5: Some generalities made about the observed activities for compounds 1-4 are hard to assure due to just four examples is a too short set of samples for a SAR analyses. Please be careful. Perhaps, more examples are required for a more reliable analysis.

Response 5:   The aim of the work was not to carry out a SAR study.  However, based in the results of EC50 from ten compounds (Six more compounds were added) may be suggested that some structural characteristics affect the antifungal activity (Pages 5, 6; lines 126-157) 

Point 6: A commercial antibiotic is required to be used as reference for comparison and analysis of the antifungal assays.

Response 6: Antifungal activity of a commercial fungicide (iprodione) was used as reference (pages 5, 6; lines 151, 152)

Point 7: The spectroscopic characterization of the obtained products should be revised. Mainly, the mass spectra require revision. This referee recommends the attaching of the NMR and mass spectra for all products as Supporting Information.

Response 7: Mass and NMR spectra of the ten synthetized compounds have been attached as supporting information.

Point 8: The procedure for calculate the reaction yields must be revised. This referee considers that it is wrong. Moreover, some yields reported in the experimental section does not matches with those reported in Table 1.

Response 8: Reaction yields were corrected and reaction yields of the new obtained compounds were added in table 1 (page 3, line 89). Also, yields were corrected in the experimental section (page 8-10; Spectroscopic data section)

Point 9: Due that there is not novelty in (synthetic and biological) methodologies, perhaps supplying several new examples and their antifungal results could increase the interest on this paper.

Response 9: Six more compounds and their antifungal activity were added (Tables 1-3, scheme 1, figures 1 and 3)

Reviewer 2 Report

The authors synthesized four 2,6-dimethoxy-4-phenyliminocyclohexa-2,5-dienone derivatives using the enzyme laccase from Trametes versicolor. The p-chlorinated compound 1 was found showing certain antifungal activity against Botrytis cinerea. The authors also reported the interferes of these compounds with conidial germination and fungal cell wall. Some major revisions are required as listed below:

1. The author's use of “benzenone” in naming the compounds is inaccurate. It is suggested that cyclohexa-2,5-dienone” be used instead of “benzenone”. Such as, the compound 1 should be named 4-(4-chlorophenylimino)-2,6-dimethoxycyclohexa-2,5-dienone”.

2. The authors should supplement the photographs of conidial germination in the manuscript.

3. The authors should add the spectroscopic data of IR spectra of the four compounds in the manuscript.

4. The authors should upload the supplementary materials containing all the spectrograms of 1H and 13C NMR, IR and MS.

Author Response

Point 1: The author's use of “benzenone” in naming the compounds is inaccurate. It is suggested that “cyclohexa-2,5-dienone” be used instead of “benzenone”. Such as, the compound 1 should be named “4-(4-chlorophenylimino)-2,6-dimethoxycyclohexa-2,5-dienone”.

Response 1: In the first version of the manuscript, the obtained compounds were called 2,6-dimethoxy-4-(phenylimino)benzenone derivatives. The compound names were corrected and now they are called 4-(phenylimino)-2,6-dimethoxycyclohexa-2,5-dienone derivatives (highlighted in yellow in the manuscript)

Point 2: The authors should supplement the photographs of conidial germination in the manuscript.

Response 2: Conidial germination results were removed from the manuscript. There are two reasons for this

1. The first four compounds had a very low effect on conidial germination

2. We added six new compounds and the effect on conidial germination was not determined. We only have results for the effect on mycelial growth (EC50, added in table 3, page 5, line 141)

Point 3: The authors should add the spectroscopic data of IR spectra of the four compounds in the manuscript.

Response 3:  The analysis of NMR and mass spectra of obtained compounds allowed us to determine the structure of these compounds. However, we attached the IR spectra of the ten compounds as supporting information

Point 4: The authors should upload the supplementary materials containing all the spectrograms of 1H and 13C NMR, IR and MS.

Response 4: 1H and 13C NMR, MS and IR spectrograms of the ten synthetized compounds have been attached as supporting information.

Reviewer 3 Report

The article „Antifungal activity against Botrytis cinerea of 2,6-dimethoxy-4-(phenylimino)benzenone derivatives” by Paulo Castro et al. describes enzyme laccase-catylyzed synthesis of 3 original phenyliminobenzenone derivatives with antifungal activities against phythopathogen Bortrytis cinerea. Although, this kind of biotechnological synthesis is more complicated than the classic organic synthesis, the number of investigated compounds is rather too low to discuss structure-activity relationship, what Authors try to do within this manuscript. In  order to extend the series for SAR analysis, they compare results of biological assays of both new obtained compounds and suitable substrates. In fact, only compound 1 (originally obtained previously)  displayed enough interesting EC50 to be promising as an antifungal agent with potential practical application.

Despite to rather poor experimental studies performed, Authors over-extended the question of spectral analysis to determine rather expected and unequivocal chemical structure of the obtained 3 compounds. They discuss 1H and 13C-NMR chemical shifts in 2 page-description, present in Table 3 and then repeat the same data in experimental section. 

Although I appreciate the interesting synthesis way, detailed analysis and  biological screening, including studies on influence of the compounds on cell wall integrity of  B. cinerea, I think this work is not enough for original article in Molecules journal.

Authors should at least extend the series of tested compounds, especially, since biological activity of all 3 new-synthesized compounds is not spectacular. Furthermore, the work is described unclear in several fragments, including style and grammar errors, e.g.:

1. line 66 –“The substrate ratio which better yields were obtained”

2. line 70 – “all substrate is available”

3. line 74 – “the yield reaction did not increase”

In my opinion, this work cannot be published within Molecules in the present state.

Author Response

Point 1: The article “Antifungal activity against Botrytis cinerea of 2,6-dimethoxy-4-(phenylimino)benzenone derivatives” by Paulo Castro et al. describes enzyme laccase-catylyzed synthesis of 3 original phenyliminobenzenone derivatives with antifungal activities against phythopathogen Botrytis cinerea. Although, this kind of biotechnological synthesis is more complicated than the classic organic synthesis, the number of investigated compounds is rather too low to discuss structure-activity relationship, what Authors try to do within this manuscript. In order to extend the series for SAR analysis, they compare results of biological assays of both new obtained compounds and suitable substrates. In fact, only compound 1 (originally obtained previously) displayed enough interesting EC50 to be promising as an antifungal agent with potential practical application.

Response 1: Six new compounds were added to the manuscript, four of them showing similar or better antifungal activities than compound 1 (now called 3a) antifungal activity was added in table 3 (page 5, line 141).

Point 2: Despite to rather poor experimental studies performed, Authors over-extended the question of spectral analysis to determine rather expected and unequivocal chemical structure of the obtained 3 compounds. They discuss 1H and 13C-NMR chemical shifts in 2 page-description, present in Table 3 and then repeat the same data in experimental section.

Response 2: Spectra analysis has been shortened, only one analysis was left in the result and discussion section of the manuscript. 1H and 13C NMR data of the ten synthetized compounds can be found in the Spectroscopic data section (page 8 -10).

Point 3: Although I appreciate the interesting synthesis way, detailed analysis and biological screening, including studies on influence of the compounds on cell wall integrity of B. cinerea, I think this work is not enough for original article in Molecules journal.

Authors should at least extend the series of tested compounds, especially, since biological activity of all 3 new-synthesized compounds is not spectacular. Furthermore, the work is described unclear in several fragments, including style and grammar errors, e.g.:

1. line 66 –“The substrate ratio which better yields were obtained”

2. line 70 – “all substrate is available”

3. line 74 – “the yield reaction did not increase”

Response 3: Six new compounds and their antifungal activity were added to the manuscript.

Also, the paragraph including lines 66 – 74 was corrected (Page 3, lines 73-81)

Several grammatical errors in the manuscript were corrected.

Round 2

Reviewer 1 Report

- Some style issues still remain in the manuscript. See the attached document.

-A sequence of steps for the formation of compounds 3 from 1 and 2 should  accompany Scheme 1.

- The reaction yield for the formation of compound 3a from the direct methodology starting with syringaldehyde must be supplied.

- For the known compounds 3a and 3f, any reported parameter should be supplied to compare with that obtained in this work.

- This referee newly encourage the authors to report correctly the MS spectroscopic data for all obtained compounds!!!.The Cl-rule is still lacking!!. MS?..EI?, intensities?. Please see the attached manuscript.

Author Response

Please find the respond in the document attached

Reviewer 2 Report

The order of groups is arranged in the order of the first letter of each group. So, 4-(phenylimino)-2,6-dimethoxycyclohexa-2,5-dienone should be named as “2,6-dimethoxy-4-(phenylimino)cyclohexa-2,5-dienone”.

Author Response

Point 1: The order of groups is arranged in the order of the first letter of each group. So, “4-(phenylimino)-2,6-dimethoxycyclohexa-2,5-dienone” should be named as “2,6-dimethoxy-4-(phenylimino)cyclohexa-2,5-dienone”.

Response 1:  Compounds names have been corrected according to the first letter of the group (highlighted in yellow). 4-(phenylimino)-2,6-dimethoxycyclohexa-2,5-dienone now is named 2,6-dimethoxy-4-(phenylimino)cyclohexa-2,5-dienone.

Reviewer 3 Report

Authors performed major revision of the manuscript and extended series of investigated compounds to improve SAR-discusssion.

The manuscript in the presence version can be published in Molecules journal. Authors are encourage to imrove minor language disadvantages  during proof correction.

Author Response

Point 1: Authors performed major revision of the manuscript and extended series of investigated compounds to improve SAR-discussion.

The manuscript in the presence version can be published in Molecules journal. Authors are encourage to improve minor language disadvantages during proof correction.

Response 1:  We appreciate your comments.